# Prospect of Prostate Cancer Treatment: Armed CAR-T or Combination Therapy

**DOI:** 10.3390/cancers14040967

**Published:** 2022-02-15

**Authors:** Yao Jiang, Weihong Wen, Fa Yang, Donghui Han, Wuhe Zhang, Weijun Qin

**Affiliations:** 1Department of Urology, First Affiliated Hospital of Air Force Military Medical University, Xi’an 710032, China; jiangyao@fmmu.edu.cn (Y.J.); yangfa@fmmu.edu.cn (F.Y.); handonghui0118@163.com (D.H.); 2Department of Medical Research, Northwestern Polytechnical University, Xi’an 710072, China; 3Department of Urology, Air Force 986 Hospital, Xi’an 710054, China; zwh74129@126.com

**Keywords:** CAR-T cell therapy, immunotherapy, prostate cancer, anti-PD-L1, checkpoint inhibitor

## Abstract

**Simple Summary:**

There is still no effective treatment for advanced prostate cancer. CAR-T therapy is a promising approach; however, many obstacles remain for the treatment of solid tumors due to the complex physical barriers and inhibitory microenvironment in solid tumors. Single CAR-T therapy has a low response rate and a high recurrence rate. With the enhancement of CAR-T itself and the gradual improvement of the immune microenvironment, the number of CAR-T weapons against tumors is increasing. This article discusses the current status and future of CAR-T therapy for prostate cancer. We believe that the enhancement and modification of CAR-T or CAR-T combined with other therapies are expected to be a breakthrough in the treatment of prostate cancer.

**Abstract:**

The incidence rate of prostate cancer is higher in male cancers. With a hidden initiation of disease and long duration, prostate cancer seriously affects men’s physical and mental health. Prostate cancer is initially androgen-dependent, and endocrine therapy can achieve good results. However, after 18–24 months of endocrine therapy, most patients eventually develop castration-resistant prostate cancer (CRPC), which becomes metastatic castration resistant prostate cancer (mCRPC) that is difficult to treat. Chimeric Antigen Receptor T cell (CAR-T) therapy is an emerging immune cell therapy that brings hope to cancer patients. CAR-T has shown considerable advantages in the treatment of hematologic tumors. However, there are still obstacles to CAR-T treatment of solid tumors because the physical barrier and the tumor microenvironment inhibit the function of CAR-T cells. In this article, we review the progress of CAR-T therapy in the treatment of prostate cancer and discuss the prospects and challenges of armed CAR-T and combined treatment strategies. At present, there are still many obstacles in the treatment of prostate cancer with CAR-T, but when these obstacles are solved, CAR-T cells can become a favorable weapon for the treatment of prostate cancer.

## 1. Introduction

According to the 2020 Global Cancer Statistics Report, the incidence rate of prostate cancer ranks second in all male malignant tumors [1]. Early stages of prostate cancer lack specific clinical symptoms, with the main manifestations being frequent urination, urgent urination, increased nocturia, and weak urine flow, which are similar to the symptoms of prostate hyperplasia. When obvious symptoms appear, the disease usually progresses to the middle and late stages. Androgen deprivation therapy (ADT) is often required for noncurative treatment, but after a period of treatment, the cancer often develops into castration resistant prostate cancer and a higher mortality rate appears [2]. For metastatic hormone sensitive prostate cancer, castration combined with abiraterone was approved by FDA in 2018, and the 3-year overall survival rate of patients has since increased from 49% to 66% [3]. In addition, based on two phase III clinical studies, castration combined with docetaxel chemotherapy is also strongly recommended. The combined regimen can prolong the overall survival and provide more benefits for patients with a high tumor load [4,5]. However, when the disease progresses to castration resistant prostate cancer (CRPC), ADT-based treatment is not so easy. Chemotherapy is an important treatment at this stage. The first FDA approved chemotherapeutic drug for the treatment of mCRPC was docetaxel, followed by cabataxel approved in 2010 for docetaxel unresponsive patients. ADT combined with chemotherapy or endocrine therapy, such as abiraterone and enzalutamide, are still the schemes recommended by some guidelines. However, the application of chemotherapeutic drugs leads to different degrees of drug resistance and many side effects [6]. Although other therapies such as RA233, PARP inhibitors and targeted radioactive ligands have certain curative effects, they cannot fundamentally solve the problem of poor prognosis of patients with CRPC. In the past decade, the immunotherapy of tumors has been greatly improved. In 2010, the first prostate cancer vaccine, Sipuleucel-T, was approved for the treatment of asymptomatic or slightly symptomatic patients with mCRPC [7]. However, although Sipuleucel-T can prolong the overall survival, it cannot affect the PSA level of patients, so it still needs systematic and comprehensive evaluation. Immune checkpoint inhibition and CAR-T cell therapy are new methods of tumor immunotherapy in recent years, which have achieved remarkable results in the treatment of many tumors. Pabolizumab has been approved by FDA for the treatment of prostate cancer, but a single treatment regimen has not shown significant efficacy [8]. On the one hand, PD-L1 may not be highly expressed in prostate cancer. On the other hand, prostate cancer is regarded as a “cold tumor”, with less T cell infiltration, and PD-1 inhibitory receptor is highly expressed on the surface of T cells, so the tumor is prone to immune escape.

Genetically engineered chimeric antigen receptor T (CAR-T) cells are another potential antitumor treatment option. Because CAR-T cells can recognize surface antigens independently of MHC restriction [9], they have better tumor-targeting and tumor-killing ability than conventional T cells. Immunotherapy with CAR-T cells has achieved tremendous success in treatment of hematological malignancies, but significant challenges exist for CAR-T treatment of solid tumors [10]. An important aspect is the immunosuppressive tumor microenvironment, which weakens T cell function, limits T cell proliferation, and impairs T cell recognition and killing of tumor cells. An essential factor that limits CAR-T in solid tumors is the activation of PD-1 by its ligand PD-L1. The combination of PD-1 and PD-L1 results in inhibition of T cell activities and suppression of T cell proliferation [11]. When antibodies were used to block this pathway, T cell activity was restored, and tumor regression was promoted [12]. CTLA-4 is another major inhibitory receptor of T cells. Clinical studies have shown that Ipilimumab has a significant effect on mCRPC patients without chemotherapy and visceral metastasis [13], proving that CTLA-4 is an effective target. Therefore, the combination therapy of CAR-T cells and the way of blocking immune checkpoints are potential ways to overcome the obstacles of existing effective treatment. In addition, the armed CAR-T or combination of other programs can improve the effectiveness and survivability of CAR-T cells. In this article, we review studies of CAR-T cells in the treatment of prostate cancer, and discuss the prospects for prostate cancer treatment by armed CAR-T and combined therapy.

## 2. Structure of CAR-T

CAR-T cells are genetically engineered T cells that express a unique fusion receptor. The receptor is composed of an extracellular domain, a hinge region, a transmembrane domain, and an intracellular signal transduction region. The extracellular domain generally consists of a single-chain fragment (scFv) that specifically recognizes tumor-associated antigens (TAAs) [14]. The specificity and affinity of scFv determine the tumor targeting of CAR-T cells. The most commonly used hinge region motifs are derived from IgG1, IgG4, IgD, and CD8 domains [15], and the size of the hinge area often affects the flexibility of scFv. According to different TAAs, reasonable selection of hinge region structure can improve the ability of CAR-T to recognize tumor antigens [16]. The transmembrane domain is usually composed of the transmembrane regions of CD3, CD8, CD28, or FCεRI, which anchor CAR structure to the T cell membrane [17]. The intracellular domains, named immunoreceptor tyrosine-based activation motifs (ITAMs), are the signal transduction and cell activation units. ITAMs are usually a TCR/CD3 ζ chain of the T cell receptor or the FcεRI γ chain of the immunoglobulin receptor. Because FcεRI γ has only one ITAM, whereas CD3 has three, the activation effect of CD3 is stronger and used more widely [18].

CAR-T has undergone five generations of structural and functional changes (Figure 1 and Table 1). The intracellular domain of the first-generation CAR contained only CD3ζ, forming the scFv-cD3ζ structure. Because of the lack of a costimulatory signal, these CAR-T cells quickly undergo apoptosis after application, and their antitumor activity is greatly restricted [19]. The second-generation CAR had an immune-costimulatory signaling molecule CD28 or 4-1 BB added to the intracellular region, forming the scFv-CD28/4-1 BB-CD3ζ structure. Compared with the first generation, the antigen specificity of the second-generation CAR-T cells was unchanged, but the proliferation and cytokine secretion capabilities were greatly improved [20]. The third generation of CARs added CD134 or CD137 as costimulatory signaling molecules on the basis of the second generation, forming scFv-CD28-CD134-CD3ζ or scFv-CD28-CD137-CD3 ζ. However, there was no clear evidence that the third-generation CAR had advantages compared with the second-generation CAR. A recent study showed that only second-generation CARs induced the expression of a constitutively phosphorylated form of CD3ζ [21]. In addition to the chimeric antigen receptor gene, the fourth-generation CAR had genes with immune regulatory functions, such as IL-12, IL-15, or other cytokines, to improve the antitumor activity of the CAR-T cells [22,23]. Universal CAR-T may become the main research object of the next generation of CAR-T, which can effectively abolish graft-versus-host disease (GVHD) by disrupting the TCR gene and/or HLA class I loci of the allogeneic T cells using gene-editing technology [24].

## 3. CAR-T Therapy and Problems

In 2013, the University of Pennsylvania reported the first case of a child with acute lymphoblastic leukemia who achieved complete remission after CAR-T therapy [25]. Subsequently, CD19-targeted CAR-T has been widely used in the treatment of hematologic tumors. In long-term follow-up, patients with ALL and NHL showed significant remission after CD19-CAR-T treatment [26,27,28]. For patients with multiple myeloma, anti-BCMA CAR-T had the highest effective rate. One study showed that the median follow-up time was 417 days, the overall response rate (ORR) was 88.2%, and the 1-year overall survival (OS) was 82.3% [29]. On the basis of these encouraging results, in 2017, the U.S. Food and Drug Administration approved two CAR-T immunotherapies, Kymriah and Yescarta, primarily for the treatment of ALL and NHL [30,31]. In 2020, with the results of the KTE-X19 CAR-T (Tecartus) therapy, the FDA approved a third CAR-T therapy for adult patients with MCL [32]. Overall, CAR-T therapy has shown ideal clinical efficacy in patients with hematologic malignancies. There are some adverse events, such as cytokine release syndrome, recurrence of disease due to immune escape of tumor antigens, reduction of blood cells, adverse reaction of central nervous system, infection, and ineffective platelet transfusion. However, these side effects are usually manageable [33].

CAR-T therapy has limited efficacy in solid tumors. Hou et al. [34] reported that the overall effectiveness of CAR-T cells in solid tumors was only 9%, with an overall response rate of 11% in hepatobiliary and pancreatic tumors, 12% in neurologic tumors, and 12% in other tumors, hence far less effective than in hematologic tumors.

There are major impediments to using CAR-T for solid tumors. Target specificity is the most important; however, no tumor-specific antigen has been found in solid tumors. Most target antigens are tumor-associated antigens that express at low levels in normal tissues, leading to the risk of off-target effects and even death [35]. Second, in the treatment of solid tumors, the primary killing effect of CAR-T cells is achieved only when the CAR-T cells migrate from the peripheral blood to the tumor site, especially to the interior of the primary tumor and other metastatic lesions [36,37,38]. However, solid tumors often have abnormal vascular beds and high levels of interstitial fibrosis that inhibit delivery of CAR-T cells or drugs to deep tumors [39,40]. More importantly, there is a complex immunosuppressive microenvironment in solid tumors. In detail, depleted T cells often express inhibitory receptors, including programmed death receptor 1 (PD-1), cytotoxic T lymphocyte-associated antigen 4 (CTLA-4), T cell immunoglobulin mucin-3 (Tim-3), and lymphocyte activation gene 3 (LAG-3). These inhibitory receptors bind their corresponding ligands to induce apoptosis of T cells by different mechanisms, thereby down-regulating the immune response [41,42,43]. Immunosuppressive factors secreted by tumor cells, Tumor-associated macrophages (TAMS) and regulatory T cell (Treg) in the tumor microenvironment, such as IL-10, IL35, and TGF-β, are key factors in T cell failure. TGF-β not only contributes to the activation of Tregs and tumor angiogenesis cytokines, but also TGF-β induces upregulation of CTLA-4 expression by Treg [44,45]. In addition, tumors consume a large amount of glucose and essential amino acids, and tumors produce many metabolites such as fatty acids and lactic acid that cause a hypoxic acidic microenvironment, which may reduce the cellular function of CTL [46].

Several methods have been used to improve CAR-T cell function in the tumor microenvironment. Leonid showed that depleted CAR-T was reactivated by application of PD-1 antibody [47]. John et al. found increased efficacy of CAR-T by using PD-1 blocking antibodies combined with HER2-CAR-T in HER2+ sarcoma cells [48]. Researchers also engineered CAR-T to secrete PD-1 scFv locally to block the PD-1/PD-L1 pathway, which not only prolonged the survival time of mice, but also avoided adverse events caused by systemic administration of PD-1 antibodies [49]. A more promising approach is the use of gene-editing technology, which enables modification of single or multiple genes. Ren et al. [50] used CRISPR-Cas9 technology to create PD-1 and CTLA-4 double knockout CAR-T cells, which improved the activity of the CAR-T cells. Later, they knocked out the TCR β2 microglobulin to achieve a still better antitumor effect [51]. However, other evidence showed that proliferation of CAR-T cells was inhibited after PD-1 receptor silencing or knock out [52]. Therefore, more experiments are needed to assess the feasibility of this scheme.

## 4. Targets of CAR-T Therapy in Prostate Cancer

CAR-T studies in prostate cancer are currently focused on preclinical studies, with a small number of phase I trials conducted to assess safety (Table 2). Finding specific targets for prostate cancer is the first step in the development of effective CAR-T therapy. The ideal tumor target should be expressed exclusively in cancer cells, and CAR-T can generate specific immune responses in tumor tissues without damaging normal tissues. Prostate-specific antigen (PSA) is the most commonly used marker for the diagnosis of prostate cancer. PSA is secreted by prostate vesicles and epithelial duct cells, and can be detected in serum normally with a concentration of less than 4 μg/mL [53]. When prostate tissue is destroyed, PSA is released into the blood through capillaries [54], However, these secreted target antigens are not suitable as CAR-T targets because they cannot be localized to target cells, so it is critical to find highly specific membrane surface antigens. Currently, three main targets of CAR-T therapy for prostate cancer research are prostate specific membrane antigen (PSMA), prostate stem cell antigen (PSCA), and epithelial cell adhesion molecule (EpCAM).

Prostate specific membrane antigen (PSMA) is one of the most common targets. The gene is located on chromosomes 11p11-12 and expressed as a 750-amino acid II type intrinsic membrane protein. PSMA is over-expressed on the membrane of prostate cancer and endothelial cells of tumor neovasculature [55]. It was also found in other normal tissues, such as salivary gland, brain, small intestine, renal tubular epithelium and breast epithelium [56]. In a mouse model of prostate cancer constructed from PC3 cells, Ma et al. [57] compared the efficacy of first-generation and second-generation CAR-T cells targeted to PSMA and normal T cells. They found that 75% (6/8) of the second-generation CAR-T group achieved complete remission, significantly superior to the first-generation CAR-T (1/8) and normal T cells (0/8). Zuccolotto et al. [58] introduced CD28 as a costimulatory molecule in PSMA-CAR to construct the second generation of CAR-T cells. In the treatment group, tumor volume gradually decreased after 1 week and tumors almost disappeared after 3 weeks. The survival time of the mice was also prolonged (SCID mice: 54 d in the control group, 74 d in the experimental group; NOD-SCID mice: 60 d in the control group, >150 d the experimental group). In addition, in order to weaken immunosuppressive factors, researchers added anti TGF-β to PSMA targeted CAR-T and found that the tumor killing ability of CAR-T was significantly improved [59]. In 2016, the results of a phase I clinical trial showed that two of the five patients with prostate cancer achieved partial remission with reduced serum PSA, and no toxicity caused by PSMA CAR-T cells was observed [60]. In 2018, Kloss et al. [61] built an inhibitor of TGF-β receptor expression PSMA-CAR-T cells, which improved the CAR-T effect; a phase I trial (ClinicalTrials.gov Identifier: NCT03089203) was then launched to evaluate CAR-T in mCRPC patients. Currently, some phase I/II clinical studies are ongoing, but no published data are available (ClinicalTrials.gov Identifier: NCT04249947, NCT04633148, NCT04429451).

Prostate stem cell antigen (PSCA) is a tumor-related antigen discovered by Reiter et al. [62] in a study of prostate cancer gene expression. The protein was named as a prostate stem cell antigen due to its 30% homology with stem cell antigen. PSCA has some functions of stem cells, such as cell self-renewal, proliferation and adhesion, and is involved in tumor genesis and development [63]. The expression rate of PSCA in normal prostate tissues is about 60–70%, and more than 90% in prostate cancer tissues [62]. Further studies have shown that PSCA expression gradually increased from normal prostate cancer, prostate intraepithelial tumor, hormone dependent, hormone independent prostate cancer, and bone metastases of prostate cancer [64,65]. Therefore, PSCA is an ideal target in advanced or metastatic diseases. Hillerdal et al. [65] constructed a third generation CAR-T cell targeting PSCA for the treatment of prostate cancer in mice. The vitro experiments showed that when CAR-T cells specifically bound to target cell PSCA, they secreted a large amount of IL-2 and interferon γ, which promoted CTL proliferation and killed tumors effectively [66]. Priceman et al. [66] confirmed the advantage of PSCA-CAR-T in the model of bone metastasis of prostate cancer, and they found from a selection of different costimulatory molecules that 4-1 BB enabled PSCA-CARs to have higher disease control ability and to exhibit better T cell persistence compared with CD28 as a costimulatory molecule [67]. Currently, two phase I/II clinical trials are underway to evaluate the efficacy and safety of CAR-T targeting PSCA in patients with advanced prostate cancer (ClinicalTrials.gov Identifier: NCT03873805, NCT02744287).

The third effective target is the epithelial cell adhesion molecule (EpCAM), also known as CD326, which belongs to the adhesion molecule family. The EpCAM gene is located on chromosome 2p21 and encodes a 40 kDa type I transmembrane glycoprotein. EpCAM functions as an epidermal cell adhesion molecule, participating in signal transduction and cell proliferation [68,69]. EpCAM is associated with oncogenesis and is strongly expressed in various types of human epithelial carcinoma, such as lung, breast, prostate, ovarian, cervical, and colorectal cancer (CRC), and the expression of EpCAM is related to the degree of disease, suggesting that it may be a promising target for cancer diagnosis and treatment [70]. Some studies suggest that EpCAM can be used as a predictor of prostate cancer; it has an important activity in CaP proliferation, invasion, metastasis, and chemo-/radio-resistance associated with the activation of the PI3K/Akt/mTOR signaling pathway [71]. Using EpCAM as TAA, researchers built EpCAM-specific chimeric antigen receptors. Although the EpCAM on PC3 cells was expressed at a low level, in the transfer model, EpCAM CAR-T still inhibited the growth of tumors and increased the survival time of mice. Thus, EpCAM may be better for high proliferation and metastasis of cancer cells [72]. However, the expression of EPCAM in prostate cancer is inconsistent. Some studies have shown that EpCAM expression has no significant correlation with Gleason score and progression after radical treatment in prostate cancer [73]. Another study confirmed that the overexpression of EpCAM was significantly associated with high Gleason grade by tissue microarray method. Therefore, the merits and demerits of EPCAM as a target for CAR-T therapy in prostate cancer need further confirmation. One clinical trial has begun to evaluate the safety and efficacy of CAR-T cells that target EpCAM in patients with EpCAM-positive cancer (ClinicalTrials.gov Identifier: NCT03013712).

**Table 2 cancers-14-00967-t002:** Study of CAR-T in the treatment of prostate cancer.

Publication Year	Country and Region	Study Type	Target	Generation	Main Outcome
2018 [66]	City of Hope, Duarte, CA, USA.	Preclinical study	PSCA	Second generation	4-1BB-containing CARs show superior T cell persistence and control of disease compared with CD28-containing CARs.
2018 [61]	Philadelphia, PA, USA	Preclinical study	PSMA	Second generation	CAR-T cells could be enhanced by the co-expression of a dominant-negative TGF-βRII (dnTGF-βRII).
2019 [74]	Tehran, Iran	Preclinical study	PSMA	Second generation	NBPII-CAR- increases the antitumor activity of CAR-T cells.
2020 [75]	Freiburg, Germany	Preclinical study	D7-based PSMA-targeting	Second generation	D7-derived CAR-T cells significantly inhibited tumor growth in combination with low-dose docetaxel.
2020 [76]	Shanghai, China	Preclinical study	IL-23 PSMA	Second generation	Duo-CAR-T cells co-expressed the IL-23mab and PSMAmAb has significant proliferation and activation effects.
2020 [77]	Xinjiang Medical University, China	Preclinical study	PSMA-CAR co-expression of ICR (an inverted chimeric cytokine receptor)	Second generation	Co-expression of ICR could significantly enhance sustained antitumor capabilities of PSMA-CAR-T cells.
2020 [78]	Shanghai, China	Preclinical study	NKG2D-CAR-T co-expression of IL7	Second generation	NKG2D-CAR-T cells performed significantly increased cytotoxicity against prostate cancer.
2021 [79]	Nanchang, Jiangxi, China	Preclinical study	B7-H3(CD276)	Second generation	B7-H3 CAR-T cells were highly cytotoxic to FIR treated PCSCs.

Abbreviations: Prostate stem cell antigen (PSCA), Prostate specific membrane antigen (PSMA), Transforming growth factor-beta receptor type II (TGF-βRII), Interleukin-23 (IL-23), Inverted chimeric cytokine Receptor (ICR), Natural Killer Group 2D (NKG2D), Interleukin-7 (IL-7), Fractionated irradiation (FIR), Prostate cancer stem cells (PCSCs).

## 5. Prospects of CAR-T for the Treatment of Prostate Cancer

### 5.1. Improve Safety Performance

CAR-T therapy for solid tumors requires identification of a specific tumor antigen. However, there is no actual tumor-specific antigen. Some tumor-associated antigens (TAA) can be selected, but they are often highly expressed in tumors and with low-level expression in some normal tissues, resulting in the destruction of normal tissue cells while killing tumor cells. For prostate cancer, although prostate-specific antigen (PSA) and prostate acid phosphatase (Pap) are highly specific, they are secretory markers and cannot be used as targets of CAR-T. PSCA and EpCAM are overexpressed antigens related to tumor invasiveness, but their expression in normal tissues leads to toxicity [80]. PSMA is weakly expressed in many organs, including the bladder, proximal tubules of the kidney, liver, esophagus, stomach, small intestine, colon, breast, and ovarian stroma [81]. Interestingly, PSMA is highly expressed in tumor neovascularization, but almost not in normal vessels. Noss et al. suggested that PSMA promoter/enhancer specifically enhances PSMA transcription in prostate cancer cell lines, but this enhancement of transcription was not found in non-prostate cell lines or prostate cell lines that did not express PSMA [82]. Therefore, PSMA is still a potential research target.

Another strategy to improve the safety of CAR-T cell therapy is to introduce the suicide gene, which can eliminate CAR-T cells when adverse toxic reactions occur. Herpes simplex virus-derived thymidine kinase is a highly immunogenic viral-derived protein that can cause cell death by blocking DNA synthesis [83]. A second system is inducible caspase 9 (iCasp9). The small molecule dimer drug AP1903 induces iCasp9, which rapidly initiates the CAR-T apoptotic pathway [84]. Di Stasi et al. [85] successfully tested the suicide effects of the iCasp9 system. Within 30 min of AP1903 injection, more than 90% of CAR-T was eliminated in four patients who had developed GVHD. In addition, it is important to balance the relationship between efficacy and toxicity in CAR-T cell therapies. Watanabe et al. [86] used the concept of a “treatment window” in pharmacological toxicology for CAR-T cell therapy to achieve the highest therapeutic benefit within the acceptable range of toxicity, which can expand and optimize the clinical use of CAR-T cells for solid tumors.

### 5.2. Enhance CAR-T Cell Homing to Tumor Site

Previous studies have shown that the infiltration degree of T lymphocyte in tumors is mostly related to better clinical prognosis [87]. Only 1–2% of T cells reinfused by adoptive therapy can really enter the depth of the tumor, resulting in a great reduction of killing efficiency [88]. Prostate cancer often has few CD8+ T cells and poor response to immunotherapy, which has also been confirmed in the TCGA public database. Therefore, it is important to know the mechanism for recruiting T cells into the depths of the tumor. In the tumor microenvironment, some chemokines and their receptors can recruit various immune cells to play a role, so CAR-T cells expressing chemokine receptors is a potential strategy. One study showed that Reed–Stemberg cells of Hodgkin’s lymphoma (HL) mainly produce CCL17 and CCL22, which preferentially attract type 2 T helper cells (Th2) and regulatory T cells (Treg) that express the TARC/MDC-specific chemokine receptor CCR4. Cd30-CAR-T cells that overexpressed CCR4 enhanced their migration to HL cells and had a better clinical response [89]. Craddock et al. [90] found that the homing ability of GD2-targeted CAR-T was enhanced by co-expression of chemokine receptor CCR2B. The neuroblastoma cell lines and primary tumor cells from six patients both secreted high levels of CCl2, had good migration, and at least a 10-fold increase in homing ability. At present, it is necessary to clarify the expression pattern of specific chemokines and their receptors in prostate cancer. Through gene modification, CAR-T cells overexpressed specific chemokine receptors, so as to enhance their homing ability and play a targeted antitumor role.

### 5.3. Nanocarriers Applied to CAR-T

The use of novel nanocarriers can improve the tumor infiltration of CAR-T. Variable domain of the heavy-chain antibody (VHH) has a monomer structure, high solubility, and high specificity [91]. Hassani et al. [74] first used camelid nanobody (VHH) to construct PSMA-targeted CAR-T cells. In addition, VHH-CAR-T cells were proliferated by nearly 60% when co-cultured with LNCaP, as compared with PSMA negative prostate cancer cell (DU-145). This study proved that CAR-T cells targeted by nano antibodies can inhibit prostate cancer. The complex process for manufacturing CAR-T cells involves a series of specialized separations, genetic modification and amplification procedures. It can only be injected into patients after all aspects of quality control reach the standard; however, the expensive price limits its application. Nanomaterials are simple and inexpensive to prepare and can also be used as natural carriers of DNA. Recent studies have shown that nanocarriers carrying CAR genes can recognize and integrate T cell genes in vivo, and their antitumor effects are similar to those of T cells programmed in vitro. These engineered T cells proliferated and differentiated into memory T cells in vivo [92]. The efficiency of CAR-T proliferation determines the persistence of tumor regression. Nanotechnology can be used to stimulate the expansion and persistence of CAR-T cells without toxicity. Li Tang’s team stimulated T cells in tumors to expand 16-fold by using nanogels that carry interleukin-15 superagonist complexes [93]. Another study attached the antigen to a bundle of carbon nanotubes, binding the complex to polymer nanoparticles containing magnetite and the T cell growth factor interleukin-2 (IL-2). Results show that carbon nanotube-polymer composite can efficiently expand the number of T cells isolated from mice [94]. This evidence shows that scientific nano material design can greatly improve the amplification efficiency of CAR-T, which is expected to gain advantages in clinical preparation technology.

### 5.4. New Types of Gene Editing

A difficult question for CAR-T applications is, “how do CAR-T cells survive in a suppressed immune microenvironment?” CRISPR/Cas9, the new gene-editing technology, has been used gradually in cellular immunotherapy because of its high efficiency and simple operation. The development of CAR-T cells by gene-editing technology will improve the therapeutic potential of CAR-T cells in the treatment of blood and solid tumors. Eyquem et al. used CRISPR/Cas9 technology to deliver CD19-specific CAR-T into the T cell receptor α constant (TRAC) locus. More potent CAR-T cells were produced and delayed the differentiation and depletion of effector T cells, thereby enhancing tumor immune rejection [95]. These findings highlight the potential of CRISPR/Cas9 genome-editing technology to advance cancer immunotherapy. Because of the important mechanism of PD-1/PD-L1 in the tumor immune microenvironment, a series of gene edits are needed for this pathway. Hu et al. [96] used CRISPR/Casp9 to disrupt the programmed cell death 1 (PD-1) gene in human primary T cells, with little effect on cell proliferation but strong enhancement of CAR-T cell cytokine production and cytotoxicity against PD-L1-expressing cancer cells in vitro. Ren et al. [51] used CRISPR/cas9 technology to eliminate the expression of PD-1 in PSCA targeted CAR-T cells. The activity and antitumor ability of CAR-T cells were enhanced during co-culture with tumor cells. Although no clinical studies have been carried out, there is no doubt that CAR-T modified by gene-editing technology is a direction worthy of exploration.

### 5.5. Combined Therapy with Other Treatment Strategies

#### 5.5.1. CAR-T Combined with Radiotherapy or Chemotherapy

Radiotherapy is a commonly used local treatment for tumors. Radiotherapy promotes the release of tumor-related antigens and pressure signals, thus triggering the regression of tumors at non-radiotherapy sites (Figure 2). More importantly, radiotherapy improves the local tumor microenvironment, enabling CAR-T cells to infiltrate into the tumor and exert effective antitumor effects [97]. Weiss et al. [98] showed that low-dose radiotherapy combined with NKG2D-CAR-T cells increased the number of CAR-T cells that reached the tumor site, increased interferon-γ secretion, improved therapeutic efficacy, and extended survival of mice. In a mouse model of pancreatic tumors, DeSelm et al. [99] found that CAR-T cells were more sensitive to tumors after low dose radiation therapy, with significantly higher CR and PR rates. These studies show that radiotherapy is an important method to reshape the tumor immune microenvironment, and radiotherapy can induce a new immune homeostasis, which is an effective synergistic method of CAR-T cell therapy.

Chemotherapy can also be used as a pre-treatment method for CAR-T cell therapy. Chemotherapy eliminates lymphocytes or negatively regulating immune cells, reshapes the microenvironment of immunotherapy, and improves the proliferation and immune activity of CAR-T cells. Several studies have shown that CAR-T sensitivity was enhanced after pretreatment with chemotherapy drugs such as fludarabine and cyclophosphamide, which enhanced the efficacy of CAR-T in ovarian cancer. Paclitaxel combined with cyclophosphamide as a pretreatment regimen followed by EGFR-CAR-T cell therapy led to improved survival in patients with advanced cholangiocarcinoma [100,101]. Alzubi et al. [75] showed that systemic intravenous CAR-T cells combined with low-dose docetaxel significantly inhibited the tumor growth compared with each treatment alone. Junghans et al. [60] used fludarabine combined with cyclophosphamide to remodel the microenvironment for PSMA targeted CAR-T cells. Results showed that 40.0% (2/5) of the patients had decreases in PR and PSA levels by 50 to 70%.

#### 5.5.2. Combination with Oncolytic viruses Therapy

Some types of tumors have low response to immunotherapy. The new idea is how to transform cold tumors into hot tumors. Oncolytic viruses (OVs) can not only selectively infect and lyse tumor cells [102], but they can also enhance tumor antigen presentation, stimulate their own immune response, and regulate the immunosuppressive microenvironment by inducing antiviral response, inflammatory response, and the production of cytokines (such as GM-CSF) [103,104]. CAR-T cell therapy combined with gene-modified OVs can significantly induce CAR-T cells to penetrate TME and improve their therapeutic effect in solid tumors [105]. In a mouse model of prostate cancer, Tanoue et al. [106] used an “all-in-one” treatment whereby HER-2-targeted CAR-T cells were combined with an oncolytic adenovirus that specifically expressed PD-L1 antibodies. In vivo and in vitro experiments showed that the PD-L1 antibody expressed by the oncolytic adenovirus effectively blocked the binding of PD-1 to PD-L1, promoted the proliferation of CAR-T cells by a factor of 1.3–2 times, and enhanced the killing effect of CAR-T cells by a factor of 2–3 times. The concept of “all-in-one” therapy provides a new idea for immunotherapy of prostate cancer.

#### 5.5.3. Combination with Photothermal Therapy (PTT)

The principle of photothermal therapy for tumors is to inject photothermal conversion materials into organisms or tumors and convert light energy into heat energy with a specific external light source to achieve “burning” of tumors. Mild hyperthermia of a tumor can reduce its dense structure and inter tissue fluid pressure, increase blood perfusion, release antigens, and promote the recruitment of immune cells. The combination of photothermal therapy and CAR-T cells can potentially increase the accumulation of these cells in solid tumors and enhance the efficacy [107]. Vascular-targeted photodynamic (VPT) therapy has been applied in localized prostate cancer. The puncture negative rate of patients with low-risk prostate cancer treated with WST11A (new photosensitizer) exceeded 80% after 6 months, which proves the effectiveness of photothermal therapy [108]. Although photothermal therapy combined with CAR-T therapy for prostate cancer has not been reported, it is undoubtedly worthy of exploration. Especially, the prostate is closer to the rectum and bladder, which is more conducive to the implementation of photothermal therapy.

#### 5.5.4. Combination with Immune Checkpoint Inhibitors

T cell suppression mediated by programmed death receptor 1 (PD-1) is associated with immune escape in solid tumors. There is considerable evidence that CAR-T cell therapy and PD-1 checkpoint blockade are the ideal combination for treatment of solid tumors [47,109]. Cherkassky et al. and Song et al. used PSMA-targeted second-generation CAR-T cells in combination with PD-1 antibodies to treat prostate cancer. They found enhanced efficacy of CAR-T, and, although the duration was short, it was also an optimized regimen that could be easily adapted to the clinic [110].

### 5.6. Dual-Target or Multi-Target CAR-T Therapy

Antigen escape is the main potential mechanism for immunotherapy evasion. Thus, CAR-T cells that targeted multiple antigens are a powerful approach to address this problem. Double-antigen-targeted CAR-T cells bind to two single antigens, thereby overcoming antigen escape and improving target antigen specificity [111]. For example, Feldmann et al. [112] developed a new generic CAR (Uni-CAR) that indirectly binds multiple TAAs through target molecules and contributes to killing tumors. The target diversity and specificity of Uni-CAR-T cells make the therapy more flexible, safer, and more effective. Kloss et al. [113] constructed a dual-targeted CAR-T cell that targets PSMA and PSCA. In a mouse model of prostate cancer, these dual-targeted CAR-T cells eradicated PSCA^+^PSMA^+^ tumors, although they had a poor response to PSCA^+^PSMA^−^ tumors. These findings suggested that dual-targeted CAR-T can better identify tumors that express both antigens. In 2000, Shah et al. [114] reported the first clinical trial of bispecific anti-CD20, anti-CD19 (LV20.19) CAR-T cells. They found that bi-specific CAR-T had low toxicity and high efficacy; more importantly, there was no loss of CD19 antigen in patients who relapsed or failed treatment, which suggested that the bi-specific CAR-T can reduce recurrence by mitigating target antigen downregulation [114]. Next, we describe some published dual-target and multi-target CAR-T modes.

#### 5.6.1. Combination of Two CAR-T Cells

A mixture of two CAR-T cells, each targeting a different antigen is shown in Figure 3A. Combined targeting of these tumor-associated antigens can counteract the escape mechanism and exhibit stronger antitumor activity compared with single CAR-Ts, such as the HER2/IL-13Rα2 combination for glioblastoma [115], CD19/CD123 combination for B-ALL [116]. In terms of cytokine secretion and cytolysis, combinatorial CAR-T cells often exhibit higher levels than the individual CAR-T cells. However, combinations of CAR-T cells may create strong immune pressure on tumor cells, which may cause both antigens to escape simultaneously. Furthermore, the use of two CAR-T cells may lead to an imbalance in the immune population [117]. Such “cocktails” have been reported in the clinic. One female patient had advanced unresectable/metastatic cholangiocarcinoma that was resistant to both radiation and chemotherapy. The patient received two cycles of EGFR-targeted CAR-T cell infusion; after 8.5 months, a plateau began to emerge because most of the tumors expressed CD133. Thus, CD133-targeted CAR-T cells were administered. Although the patient achieved 4.5 months of PR, it is important to note that both CAR-T cell injections caused acute adverse reactions, such as acute subcutaneous bleeding [118]. Feng et al. [118] used a combination of Meso-CAR-T and CD19-CAR-T in metastatic pancreatic cancer; simultaneous delivery of Meso-CAR-T and CD19-CAR-T cells in PDAC patients was found to be safe, but with limited clinical activity. CD19-CAR-T cells proliferated well and consumed normal B cells, but Meso-CART cells showed depletion. This depletion may have been related to the difference of antigens in the environment, showing different expansion ability [119].

#### 5.6.2. Bicistronic CAR-T Cells

Figure 3B shows the design to co-express two independent CAR structures in the same cell. Because of tumor heterogeneity, there is no single specific tumor antigen. Multi-antigen targeting strategies may counteract antigen escape. Hegde et al. [115] designed a bispecific CAR-T targeting HER2 and IL13Rα2. Compared with single-target CAR-T cells and combined CAR-T cells, the bispecific CAR-T cells were more effective in preventing antigen escape and enhancing their antitumor efficacy. The CD19/CD123 bispecific CAR-T also showed similar advantages, with better antitumor activity and longer persistence. In xenotransplantation models, the bispecific CAR-T was better at preventing antigen loss and recurrence [116]. De Larrea et al. [120] compared the effects of tandem, bicistronic, and combined CAR-T targeting BCMA and GPRC5D. The bicistronic CAR-T had the best affinity, with a better killing efficacy and better survival in mice, followed by the combination of the CAR-T targeting BCMA and GPRC5D.

#### 5.6.3. Tandem Bispecific CAR-T Cells

Two different scFvs were designed to be strung together in a single T cell to elicit a different response to each of the two homologous antigens (Figure 3C). When one of the homologous antigens escapes, CAR-T cells still retains its killing activity. For relapsed or refractory B cell acute lymphoblastic leukemia, the most used target in hematologic tumors is CD19 combined with CD20 or CD22. In addition to low toxicity and high efficacy, no loss of CD19 antigen was observed in patients who relapsed or experienced treatment failure, which suggested that bispecific CAR-T could be used to reduce relapse [114,121,122,123,124]. In solid tumor studies, most tandem bispecific CAR-T cells have better antitumor activity and longer survival than single-target CAR-T cells. Because of the presence of inhibitory factors in the tumor microenvironment, such as PD-1/PD-L1, CTLA4, and LAG3, PD-L1 is highly expressed in most solid tumors and often causes attenuation of CAR-T cells. Researchers have tried to target PD-1 or PD-L1 to increase the sensitivity of CAR-T and weaken the inhibition tendency caused by the tumor microenvironment. Several studies have confirmed this idea [125,126,127,128].

#### 5.6.4. Tri-Specific CAR-T Cells

Three CARs on the same engineered T cell are expressed and specifically recognize three tumor-associated antigens as shown in Figure 3D. The powerful antitumor activity may be due to enhanced transduction activation, expanded tumor antigen coverage, and strong immune synapse formation. Tri-specific CAR-T therapy is more effective in preventing tumor recurrence and has a longer lasting antitumor effect after one antigen has been lost [129]. Researchers have designed and validated tri-specific CAR-T cells that simultaneously targeted HER2, IL13Rα2, and EPHA2; this system achieved nearly 100% clearance of GBM tumor cells [130]. To overcome immunosuppression effectively, an engineered T cell was redirected to recognize the prostate stem cell antigen and immunosuppressive cytokines, including TGF-β and IL-4. These three signals worked together to initiate T cell activation and produce a lasting effect, achieving safe, selective cytolysis [131]. The design of multi-target CAR-T needs to be improved. The most important item is to find suitable matching targets and design inhibitory signals to balance the function of CAR-T.

## 6. Conclusions

In Europe and the United States, prostate cancer is the male malignant tumor with the highest incidence. Endocrine therapy is still the basis of prostate cancer treatment, but when the hormone sensitive prostate cancer develops into castration resistant prostate cancer, the effect of treatment significantly decreases, and the mortality increases. Immunotherapy is a new cancer treatment method emerging in recent years, especially the successful application of CAR-T cells in hematological tumors, which brings more hope to patients. However, CAR-T still faces several difficulties in the treatment of prostate cancer. Although some preclinical studies have proved the effectiveness of PSMA-CAR-T in the treatment of prostate cancer, most of them were verified in SCID mice or cell lines. Once injected into a body with a normal immune system, CAR-T cells may face many obstacles. Some studies have found that there is less T cell infiltration and less PD-L1 expression in the tumor microenvironment of prostate cancer, resulting in a limited response to immunotherapy. CAR-T cell adoptive therapy seems to solve the targeting and source of T cells, but these cells often weaken and lose functions due to the complex tumor inhibitory microenvironment. A large body of evidence has shown the existence of inhibitory factors in the tumor microenvironment of prostate cancer. Immune infiltrating cells include cancer-related fibroblasts (CAF), bone marrow-derived suppressor cells (MDSC), tumor-associated macrophages (TAMs) and Tregs; tumor-related mediators include CTLA-4, PD-1/PD-L1, TGF beta, and adenosine, which directly or indirectly lead to the failure of CAR-T treatment. As described in our review, combination therapy is a commonly used strategy. By reshaping the immune microenvironment, which may change the cold tumor of prostate cancer into a hot tumor and improve the immunogenicity, it can improve the therapeutic activity of CAR-T cells. In addition, different combination or sequential schemes can produce different synergistic effects, and we should also consider how to balance the advantages and disadvantages of combination therapy. By optimizing the structure of CAR-T, it can integrate multiple functions. Scientific and intelligent multi-target design and reasonable target combinations under different modes need to be explored, which should not only improve the coverage of tumor antigen recognition and the sensitivity of tumor recognition, but also avoid new forms of off-target effects.

In conclusion, the characteristics of prostate cancer and tumor microenvironment pattern are important obstacles to CAR-T treatment. Through enhanced modification of CAR-T and exploration of strong alliance models, we still believe that CAR-T treatment can become a powerful weapon for the treatment of prostate cancer in the future.

## Figures and Tables

**Figure 1 cancers-14-00967-f001:**
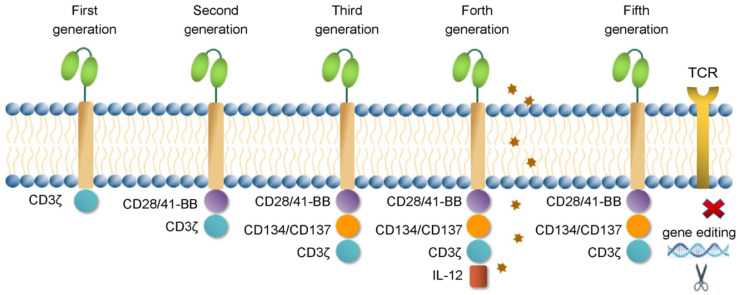
Structures of CAR-T.

**Figure 2 cancers-14-00967-f002:**
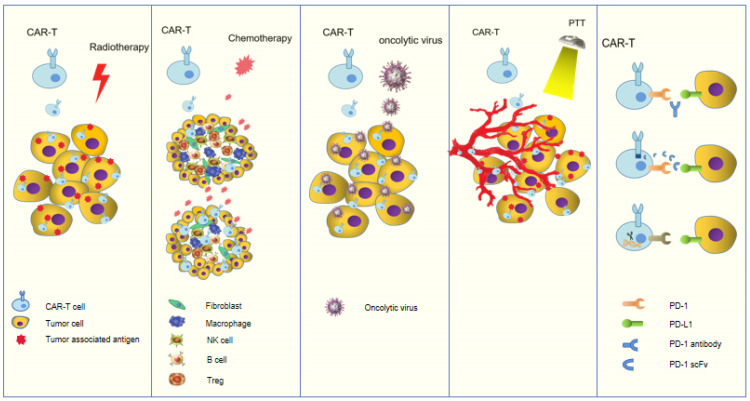
Several forms of CAR-T combination therapy.

**Figure 3 cancers-14-00967-f003:**
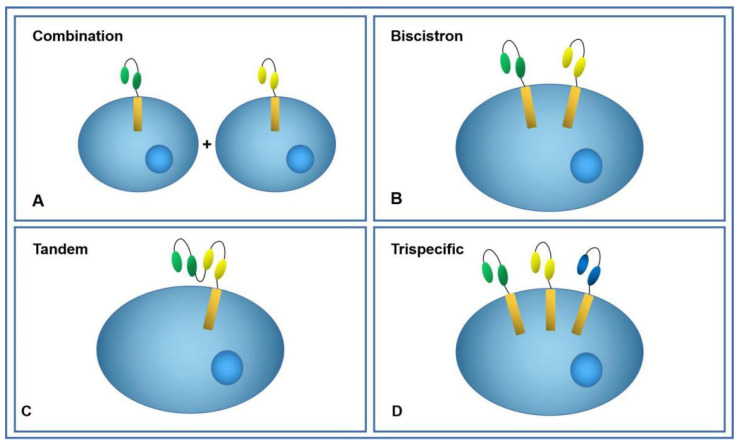
Multi-antigen targeted CAR-T cell. (**A**) Mixture of two independent CAR T cells. (**B**) Two different CAR structures were co-expressed in the same cell. (**C**) Two different scFvs were designed to be strung together in a single T cell. (**D**) Three different CAR structures were co-expressed in the same cell.

**Table 1 cancers-14-00967-t001:** Intracellular modification and function of CAR-T.

Generation	CAR-T Intracellular Modification	Efficacy
First	CD3ζ	T cells activate in vitro and have the lethality of conventional T cells but fail to proliferate and survive.
Second	CD28-CD3ζ	With the addition of a costimulatory molecule, the survival time in vivo is extended, and the proliferation ability and the killing toxicity is increased.
Third	CD28-CDl34-CD3ζ or CD28-CDl37-CD3ζ	Addition of two different costimulatory molecules improves the increment ability and killing toxicity.
Forth	Add a suicide gene or CAR-T secretes specific cytokines	Addition of suicide genes or release of immune factors refines control.
Fifth	Universal CAR-T	No individual restrictions, can be large-scale production and treatment.

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
