# Peer review of "Prospect of Prostate Cancer Treatment: Armed CAR-T or Combination Therapy"

_cancers, 2022, doi:10.3390/cancers14040967_

Round 1
Reviewer 1 Report
The main topic of the manuscript is really interesting, but there are notable shortcomings:
1) The introduction is too basic. It is necessary to rewrite the section in a more updated way. In particular:
a) Why do the authors never mention chemotherapy as a treatment option for prostate cancer? It seems a more purely urological but not oncological point of view.
b) The authors distinguish between early stage and castration resistance, but generally they do not deal with metastatic disease and do not mention any of the options available today, i.e. chemotherapy (Taxotere, Cabazitaxel), hormone therapy (abiraterone, enzalutamide and the other new drugs) or Radiopharmaceuticals (Radium-223, Lutetium); Sipuleucel-T should also be mentioned.
c) “During the preceding decade, immunotherapy strategies have greatly improved”. This is a generic phrase, which concerns other types of cancer, for which immunotherapy has already been approved. It must be explained that this is not the case with prostate cancer.
d) “Cancer immunotherapy was designated as breakthrough of the year already in 2013. These advances include monoclonal antibodies, bispecific antibodies, tumor vaccines, inhibition of immune checkpoints, and CAR-T cells. Chimeric antigen receptor T (CAR-T) cells are…” Since the authors also mention other immunotherapy approaches, explain why they do not elaborate on the topic. It would be better if the authors deleted most of the article, which is not about prostate cancer, and to enter data on immunotherapy studies in the tumor they are dealing with.
2) The manuscript in many parts reads like a general CAR review for solid tumors and the current roadblocks of CAR T cell therapy; for example, the pros and cons of CAR targets expressed on prostate cancer are not discussed in detail.
3) Page 2 lines 6-8: “A key factor that limits CAR-T in solid tumors is activation of PD-1 by its ligand PD-L1. The combination of PD-1 and PD-L1 results in inhibition of T cell activities and suppression of T cell proliferation”. This sentence is too simplistic.
a) If it were the only reason for the ineffectiveness of the treatment, it should be explained why the studies that tested immune checkpoint inhibitors did not show significant results against prostate cancer.
b) The authors should explain what differences exist between Immune checkpoint inhibitors and CARs, and what drove researchers towards the latter.
4) “The receptor is composed of an extracellular domain, a transmembrane domain, and an intracellular signal transduction region”. The hinge-region is equally important; it is true that the authors mention it later, but it should also be better if inserted in this sentence.
5) The paragraph 3 “Clinical data of CAR-T treatment” (including table 2) is too long and a bit off topic. Please remove table 2.
6) Page 5 lines 20-22 “Treatment for prostate cancer is based on endocrine therapy […] However, the disease often evolves into castration-resistant prostate cancer, and there is currently no recognized treatment for this stage of prostate cancer”. Are you sure?
7) The paragraphs 5.2 “Enhance CAR-T cell homing to tumor site”, 5.5.2 “Combination with oncolytic adenovirus therapy” and 5.5.3 “Combination with photothermal therapy” are a bit off topic. Please modify
8) The authors cite only a few studies on CAR-T in prostate cancer. Please improve this aspect, even by inserting a specific table. An example of this could be the following article (DOI: 10.1186/s40425-019-0741-7)
9) Conclusion section:
a) Page 12 lines 2-3 “The first-line treatment of advanced prostate cancer is still endocrine therapy”. This is still valid for the non-advanced stages. Please explain that chemotherapy may also be necessary in the hormone-sensitive phase in case of high tumor burden disease at diagnosis
b) Most of the paragraph is not prostate-specific. Please rewrite that section
10) Page 2 line 32 “CARS” please modify
11) Page 6 line 8 “Prostate Stem Cell Antigen” please use acronyms
Author Response
Point 1: The introduction is too basic. It is necessary to rewrite the section in a more updated way. In particular:
- Why do the authors never mention chemotherapy as a treatment option for prostate cancer? It seems a more purely urological but not oncological point of view.
- b) The authors distinguish between early stage and castration resistance, but generally they do not deal with metastatic disease and do not mention any of the options available today, i.e. chemotherapy (Taxotere, Cabazitaxel), hormone therapy (abiraterone, enzalutamide and the other new drugs) or Radiopharmaceuticals (Radium-223, Lutetium); Sipuleucel-T should also be mentioned.
- c) “During the preceding decade, immunotherapy strategies have greatly improved”. This is a generic phrase, which concerns other types of cancer, for which immunotherapy has already been approved. It must be explained that this is not the case with prostate cancer.
- d) “Cancer immunotherapy was designated as breakthrough of the year already in 2013. These advances include monoclonal antibodies, bispecific antibodies, tumor vaccines, inhibition of immune checkpoints, and CAR-T cells. Chimeric antigen receptor T (CAR-T) cells are…” Since the authors also mention other immunotherapy approaches, explain why they do not elaborate on the topic. It would be better if the authors deleted most of the article, which is not about prostate cancer, and to enter data on immunotherapy studies in the tumor they are dealing with.
Response 1:
a)Since the topic of this article is CAR T therapy based on prostate cancer, chemotherapy is not emphasized. Instead, in section 5.5.1 of this article, we introduce the effects of chemotherapy and CAR T combination. In fact, chemotherapy drugs such as docetaxel and cabattaxel can treat hormone-sensitive prostate cancer. Can significantly improve the overall survival of patients with high-load metastatic prostate cancer. For castration-resistant prostate cancer, the progression of cancer-related symptoms can be delayed, but the use of docetaxel is often limited by severe adverse reactions, and patients can achieve a higher survival benefit when appropriate in combination with other drugs. In revising the manuscript, I will add these contents appropriately.
b):The introduction here is really quite simple, without distinction and detailed explanation. Advanced prostate cancer includes locally advanced prostate cancer, metastatic hormone sensitive prostate cancer (MHSPC) and castration-resistant prostate cancer (CRPC). The former two are based on endocrine therapy and can effectively prolong the survival of patients, while the latter is still difficult to treat, with increased mortality, and there is no unified standard treatment plan. In addition to hormone therapy, there are chemotherapy, radium, vaccines, monoclonal antibodies, or combination therapy, sequential therapy, etc. However, there is still no high-level evidence to recommend different treatment regimens.
C):It will be further explained in the revised manuscript
D):The purpose of this paper is to illustrate the prospects of CAR T in the treatment of prostate cancer. Although there are many research directions of immunotherapy in clinical practice, and there are many gratifying effects, but it is not consistent with the theme of this paper, so it is only briefly described. Although car-T has not yet shown significant therapeutic responses or approved drugs in solid tumors such as prostate cancer, it is still worth exploring as a strategy and may have broad application prospects once a breakthrough is made.
Point 2: The manuscript in many parts reads like a general CAR review for solid tumors and the current roadblocks of CAR T cell therapy; for example, the pros and cons of CAR targets expressed on prostate cancer are not discussed in detail.
Response2: In the previous part of this paper , some basic knowledge of CAR-T was introduced , and some practical cases were used to explain the dysfunction of CART in solid tumors . Therefore , it may be improved to seek breakthroughs from these perspectives . Prostate cancer is one of the solid tumors , and the low response to T cells may have similar causes as other solid tumors . Therefore , solid tumors are described in some parts of this paper .
In addition, although PSMA has a high specificity, it is not absolute specificity. The low expression of PSMA in some normal tissues is associated with a risk of off-target effects, which was described in the original manuscript, but was removed at the time of submission because it is a common phenomenon in solid tumors and has been described in many articles. I will recap these contents in the article.
Point 3: Page 2 lines 6-8: “A key factor that limits CAR-T in solid tumors is activation of PD-1 by its ligand PD-L1. The combination of PD-1 and PD-L1 results in inhibition of T cell activities and suppression of T cell proliferation”. This sentence is too simplistic.
- a) If it were the only reason for the ineffectiveness of the treatment, it should be explained why the studies that tested immune checkpoint inhibitors did not show significant results against prostate cancer.
- b) The authors should explain what differences exist between Immune checkpoint inhibitors and CARs, and what drove researchers towards the latter.
Response 3:
This is indeed a little simple. Immune checkpoints such as PD-1 may be effective when PD-L1 is highly expressed in tumors, but PD-L1 expression may not be common in prostate cancer, so the effect of PD-1 antibody alone may not be significant. Prostate cancer patients who respond to immune checkpoint inhibitors generally show longer-lasting remissions, suggesting that the role of ICIs in prostate cancer warrants further investigation. Another point to note is that there are many kinds of immune checkpoints, such as CTLA-4, LAG3, TIM3, TIGIT, etc., which are inhibitory receptors of T cells. Studies have shown that the combination of PD-1 antibody and CTLA-4 antibody is more effective in prostate cancer, confirming the existence of other immune checkpoints in prostate cancer, not just the PD-1/PDL1 pathway. In this paper, PD-1 is mentioned a lot because it is the most studied immune checkpoint, but it should be noted that PD-1 is not the only program of immune checkpoint. I will revise the explanation in the revision manuscript.
Point 4: “The receptor is composed of an extracellular domain, a transmembrane domain, and an intracellular signal transduction region”. The hinge-region is equally important; it is true that the authors mention it later, but it should also be better if inserted in this sentence.
Response 4:I have inserted this sentence in my revised manuscript.
Point 5:The paragraph 3 “Clinical data of CAR-T treatment” (including table 2) is too long and a bit off topic. Please remove table 2.
Response 5:This paragraph was originally intended to explain the difference in the therapeutic effect of CAR T in hematological tumors and solid tumors, but it is indeed a little lengthy and somewhat beside the point as you said, so it will be deleted according to the comments of the review.
Point 6:age 5 lines 20-22 “Treatment for prostate cancer is based on endocrine therapy […] However, the disease often evolves into castration-resistant prostate cancer, and there is currently no recognized treatment for this stage of prostate cancer”. Are you sure?
Response 6: My description here is not accurate. First, almost all patients with advanced prostate cancer will eventually progress to Castration resistant prostate cancer (CRPC) and become resistant to first-line ADT and other hormonal therapies. CRPC remains a highly fatal disease despite the availability of multiple therapies, including hormone therapy, chemotherapy, radiotherapy, immunotherapy, and combination and sequential therapy. Docetaxel is the most widely used chemotherapy regimen in clinical practice. Although alternative therapies such as abiraterone acetate and Sipuleucel-T are not ideal for patients with docetaxel resistance, the treatment effect is not ideal. Therefore, it is an important direction of research to develop personalized treatment drugs for different patients and improve their survival efficiency.
Point 7: The paragraphs 5.2 “Enhance CAR-T cell homing to tumor site”, 5.5.2 “Combination with oncolytic adenovirus therapy” and 5.5.3 “Combination with photothermal therapy” are a bit off topic. Please modify
Response 7: The first paragraph of Section 5.2 describes the local application of CAR-T, which is indeed redundant and does not belong to a strategy of car-T modification. The improvement of chemotactic performance of CAR-T by expressing chemokine receptor in the second paragraph belongs to the self-modification of CAR-T. So whether this part should be retained, please review experts to decide.
Explanation of 5.5.2 and 5.5.3: Oncolytic viruses can not only specifically target tumors, but also transport specific genes, so the combination of the two can increase car-T targeting performance and killing efficiency to a certain extent.
Photothermal therapy not only has a direct tumor killing effect, but also can promote the exposure of tumor antigens and the expansion of blood vessels and tumor stroma, which is conducive to the entry of antibodies or small molecule drugs or car-t into the tumor. Therefore, this combined effect is also a potential treatment strategy. For prostate cancer, it can be preliminarily diagnosed through digital rectal examination, so combined photothermal therapy has congenital advantages. For these two parts, some modifications have been made in the text. Please consider whether to delete this part.
Point 8:The authors cite only a few studies on CAR-T in prostate cancer. Please improve this aspect, even by inserting a specific table. An example of this could be the following article (DOI: 10.1186/s40425-019-0741-7)
Response 8: Thanks to the evaluation experts for their valuable advice. I have added the table as suggested, but for the sake of distinction, I listed only the relevant studies in the last 5 years.
Point 9: Conclusion section:
- Page 12 lines 2-3 “The first-line treatment of advanced prostate cancer is still endocrine therapy”. This is still valid for the non-advanced stages. Please explain that chemotherapy may also be necessary in the hormone-sensitive phase in case of high tumor burden disease at diagnosis
Response 9a:This sentence is not rigorous, and the final conclusion has been rewritten.
- Most of the paragraph is not prostate-specific. Please rewrite that section
Response 9b: Rewritten as required.
Point 10: Page 2 line 32 “CARS” please modify
Response 10: It has been modified as required.
11) Page 6 line 8 “Prostate Stem Cell Antigen” please use acronyms
Response 11: It has been modified as required.
Reviewer 2 Report
The manuscript by Jiang et al constitutes a very informative and comprehensive review of both the existing and emerging modalities using CAR-T cells. All figures and tables are also carefully designed and highly informative. The authors discuss the prospects and challenges of armed CAR-T strategies and of CAR-T strategies combined with other therapeutic modalities, in the treatment of several types of cancer, with special emphasis on prostate cancer (Pca). Importantly, they include recent modifications on CAR-T cell technology that have the potential to overcome existing therapeutic limitations. The authors should however consider the following recommendations prior to publishing the review article:
- In line 112 the authors say: “For patients with multiple 111 myeloma, anti-BCMA CAR-T had the highest effective rate, almost exceeding 80%.” Effective rate in which sense? Remission? Survival? Please clarify…
- Lines 212—227: since PSCA is expressed in 50-60% of normal tissue, how can we justify its usefulness in CAR-T cell therapy, considering that the authors have already discussed the existence of off-target effects? I believe that the authors need to be more specific…for example: because PSCA expression increases with tumor progression and can even be detected in the bone metastases of PCa patients, it may be used as a molecular target in CAR-T cell therapy in patients with advanced disease (as opposed to patients with localized disease). In the same sense the authors should somehow justify the usefulness of EpCAM in CAR-T therapy (somewhere along lines 236-245), especially since they mention that overexpression in healthy tissue may cause CAR-T-related toxicity.
- Lines 226-227: I could not find the second clinical trial mentioned in the manuscript (NCT0274428), at least not in relation to prostate cancer…the only reference I could find (PMID: 28123893) links the particular trial to gastrointestinal tumors and hepatic carcinoma…so I suggest that they remove the particular trial from the manuscript, unless somehow justified. I would also like to suggest the inclusion of a reference for trial number NCT03873805: Dorff, T. B., et al (2020). A phase I study to evaluate PSCA-targeting chimeric antigen receptor (CAR)-T cells for patients with PSCA+ metastatic castration-resistant prostate cancer (mCRPC). Journal of Clinical Oncology, 38.
(https://ascopubs.org/doi/abs/10.1200/JCO.2020.38.6_suppl.TPS250).
- In section 5.5.2 you may include a sentence or two on the fact that oncolytic viruses release TAAs into the circulation, or at least into the microenvironment surrounding the tumor and may therefore aid in the recognition of these TAAs by the CAR-T cells and in the induction of an effective anti-tumor immune response.
- The Conclusions section is mostly a repetition of the Introduction. I suggest that this section is re-written with emphasis given on the promising facts / research / clinical findings of CAR-T cell therapy (including the various modifications and combinations) regarding prostate cancer treatment. In this respect, it has to be acknowledged that most of the available data retaining to CAR-T cell therapy in prostate cancer (and other types of cancer) are either in the pre-clinical (mostly) or early clinical phase, and most data come from experiments in cell lines and/or animals.
- Line 500: the authors write “These seem to be two extremes”. Which seem to be two extremes? Blood versus solid tumors? Please clarify.
- Finally, there are quite a few significant grammatical errors throughout the manuscript, and overall I would say that it does not adhere to good english standards. A few prominent examples are included in lines 16, 18, 29, 35, 41, 48, 56, 57, 69-71, 91, 92, 139-140, 164, 203-206, 236, 241, 252, 256-257, 260-261, 280-282, 291-292, 302, 320, 338, 341, 366, 382, 412-414, 446, 513-516. I therefore recommend that the authors seek editorial help (language editing) before submitting a revision for re-evaluation.
Author Response
Point 1: In line 112 the authors say: “For patients with multiple 111 myeloma, anti-BCMA CAR-T had the highest effective rate, almost exceeding 80%.” Effective rate in which sense? Remission? Survival? Please clarify…
Response 1: Specifically, the median follow-up time was 417 days, ORR was 88.2%, 1-year overall survival (OS) was 82.3%, and 1-year progression-free survival (PFS) was 52.9%.
Point 2:Lines 212—227: since PSCA is expressed in 50-60% of normal tissue, how can we justify its usefulness in CAR-T cell therapy, considering that the authors have already discussed the existence of off-target effects? I believe that the authors need to be more specific…for example: because PSCA expression increases with tumor progression and can even be detected in the bone metastases of PCa patients, it may be used as a molecular target in CAR-T cell therapy in patients with advanced disease (as opposed to patients with localized disease). In the same sense the authors should somehow justify the usefulness of EpCAM in CAR-T therapy (somewhere along lines 236-245), especially since they mention that overexpression in healthy tissue may cause CAR-T-related toxicity.
Response 2:The comments of the review are very reasonable and have been revised as suggested.
Point 3:Lines 226-227: I could not find the second clinical trial mentioned in the manuscript (NCT0274428), at least not in relation to prostate cancer…the only reference I could find (PMID: 28123893) links the particular trial to gastrointestinal tumors and hepatic carcinoma…so I suggest that they remove the particular trial from the manuscript, unless somehow justified. I would also like to suggest the inclusion of a reference for trial number NCT03873805: Dorff, T. B., et al (2020). A phase I study to evaluate PSCA-targeting chimeric antigen receptor (CAR)-T cells for patients with PSCA+ metastatic castration-resistant prostate cancer (mCRPC). Journal of Clinical Oncology, 38.
(https://ascopubs.org/doi/abs/10.1200/JCO.2020.38.6_suppl.TPS250).
Response 3:Upon examination, it was found to be an error, which has been removed based on the review expert opinion and another clinical study evaluating PSCA has been added.
Point 4:In section 5.5.2 you may include a sentence or two on the fact that oncolytic viruses release TAAs into the circulation, or at least into the microenvironment surrounding the tumor and may therefore aid in the recognition of these TAAs by the CAR-T cells and in the induction of an effective anti-tumor immune response.
Response 4:It has been amended in the relevant paragraph.
Point 5:The Conclusions section is mostly a repetition of the Introduction. I suggest that this section is re-written with emphasis given on the promising facts / research / clinical findings of CAR-T cell therapy (including the various modifications and combinations) regarding prostate cancer treatment. In this respect, it has to be acknowledged that most of the available data retaining to CAR-T cell therapy in prostate cancer (and other types of cancer) are either in the pre-clinical (mostly) or early clinical phase, and most data come from experiments in cell lines and/or animals.
Response 5:Another reviewer also mentioned this problem. After careful reading, I found that I did not summarize the article and was a little off topic, so I have rewritten the last part.
Point 6:Line 500: the authors write “These seem to be two extremes”. Which seem to be two extremes? Blood versus solid tumors? Please clarify.
Response 6:The original meaning was that CAR-T produced significantly different effects in hematologic and solid tumors, but the use of "extreme" was inappropriate and has been revised.
Point 7:Finally, there are quite a few significant grammatical errors throughout the manuscript, and overall I would say that it does not adhere to good english standards. A few prominent examples are included in lines 16, 18, 29, 35, 41, 48, 56, 57, 69-71, 91, 92, 139-140, 164, 203-206, 236, 241, 252, 256-257, 260-261, 280-282, 291-292, 302, 320, 338, 341, 366, 382, 412-414, 446, 513-516. I therefore recommend that the authors seek editorial help (language editing) before submitting a revision for re-evaluation.
Response 7:I have carefully checked the points you pointed out and modified the grammar. Please review it again in the revised manuscript.
Reviewer 3 Report
This review has summarized the progress of CAR-T therapy in the treatment of prostate cancer and discussed the prospects and challenges of armed CAR-T and combined treatment strategies. However, there are some room to improve this review to show that this one is more advanced and more comprehensive than previous literatures. For examples, one review has stated the challenges and Prospects of Chimeric Antigen Receptor T-cell Therapy for Metastatic Prostate Cancer (Eur Urol. 2020 Mar;77(3):299-308.), and another has addressed the potential of CAR T cell therapy for prostate cancer (Nat Rev Urol. 2021 Sep;18(9):556-571), and so do others. The minor issue is that Figure 1 “structures of CAR-T” is very similar to previous publications, the related references should be cited.
Author Response
Thank you for your valuable comments. I will continue to improve my manuscript according to the comments of the reviewers. We drew Figure 1 with simple software that is somewhat similar to the Figures in some articles that describe the CAR structure. We didn't quote images from other articles, so should we delete or quote pictures from other articles?
Round 2
Reviewer 1 Report
The article has been greatly improved, and is now ready for publication. Pay attention to some typos (eg. RA233 instead of RA223)
Reviewer 3 Report
the revision has been significantly improved.